# Antibacterial Activity and Biodegradation of Cellulose Fiber Blends with Incorporated ZnO

**DOI:** 10.3390/ma12203399

**Published:** 2019-10-17

**Authors:** Domen Malis, Barbka Jeršek, Brigita Tomšič, Danaja Štular, Barbara Golja, Gregor Kapun, Barbara Simončič

**Affiliations:** 1Biotechnical Faculty, University of Ljubljana, Jamnikarjeva 101, 1000 Ljubljana, Slovenia; malis.domen@gmail.com (D.M.); Barbka.Jersek@bf.uni-lj.si (B.J.); 2Faculty of natural Sciences and Engineering, University of Ljubljana, Aškerčeva 12, 1000 Ljubljana, Slovenia; brigita.tomsic@ntf.uni-lj.si (B.T.); danaja.stular@ntf.uni-lj.si (D.Š.); barbara.golja@ntf.uni-lj.si (B.G.); 3National Institute of Chemistry, Hajdrihova 16, 1000 Ljubljana, Slovenia; gregor.kapun@ki.si

**Keywords:** fiber blends, cellulose fibers, ZnO, antibacterial activity, biodegradation, soil contamination

## Abstract

This research aimed to study the influence of lyocell with incorporated ZnO (CLY) for antibacterial activity and biodegradation of fiber blends composed of viscose (CV), flax (LI), and CLY. Fiber blended samples with an increased weight fraction of CLY fibers were composed, and single CLY, CV and LI fibers were also used for comparison. Antibacterial activity was determined for the Gram-negative *Escherichia coli* and the Gram-positive *Staphylococcus aureus* bacteria. The biodegradation of fiber blends was investigated by the soil burial test. The results show that the single CLY fibers exhibited high antimicrobial activity against both *E. coli* and *S. aureus* bacteria and that the presence of LI fibers in the blended samples did not significantly affect antibacterial activity against *E. coli*, but drastically decreased the antibacterial activity against *S. aureus*. LI fibers strongly promoted the growth of *S. aureus* and, consequently, impaired the antimicrobial performance of ZnO against this bacterium. The presence of CLY fibers slowed down, but did not prevent, the biodegradation process of the fiber blends, even at the highest ZnO concentration. The soil that was in contact with the fiber blended samples during their burial was not contaminated to such an extent as to affect the growth of sprouts, confirming the sustainability of the fiber blends.

## 1. Introduction

Antimicrobial activity is one of the most important functional properties of cellulose fibers since these fibers are very easily attacked by both bacteria and fungi [1,2,3,4]. Excessive growth of microorganisms on the fibers can cause health and hygiene problems for the textile wearer, on the one hand, as well as lower the value of the textile product when in use, on the other [2]. However, when fabricating antimicrobial cellulose fibers, a greater risk appears to be that the incorporation of an antimicrobial agent may destroy the fiber’s biodegradability, which represents a prominent characteristic and significant advantage of cellulose fibers from an ecological point of view. Namely, cellulose fibers are known as environmentally-friendly fibers because they are readily degraded by soil microorganisms [5]. Therefore, the creation of antimicrobial activity while maintaining the fiber’s biodegradability remains a major challenge, which requires the careful selection of an antimicrobial agent and its concentration.

Among antimicrobial agents, ZnO has attracted a lot of attention due to the general perception that it is a safe, environmentally friendly, and biocompatible substance [6,7,8]. The antimicrobial activity of ZnO is directly related to its photocatalytic performance under UV irradiation, in which reactive oxygen species (ROS) are generated [9,10,11,12,13]. It is assumed that the generation of $H_{2}O_{2}$ and •OH, in particular, is crucial for the efficiency of ZnO as an antimicrobial agent. In addition to ROS, ${\mathrm{Zn}}^{2+}$ cations released from ZnO dissolution as well as ZnO nanoparticles (NPs) contribute to the antimicrobial activity. It is believed that ROS can easily incorporate into the bacterial cell membrane, causing the disruption of cellular components via oxidative stress. ${\mathrm{Zn}}^{2+}$ cations can penetrate the bacterial cell, where they inhibit its action. If ZnO is present in the form of NPs, the direct contact of ZnO NPs with the bacterial cell wall results in the destruction of cell integrity. Accordingly, the mechanism of the antimicrobial activity of ZnO is strongly influenced by ZnO particle size and crystal structure as well as the environmental conditions. It is worth noting that under dark conditions, which apply for the biodegradation process, the antimicrobial activity of ZnO is assumed to be mainly attributed to ${\mathrm{Zn}}^{2+}$ cations and ZnO NPs and less related to ROS, since the generation of these species is conditioned by UV irradiation [13].

Our research aims to investigate the influence of ZnO on the antibacterial activity of cellulose fibers as well as their biodegradation. For this purpose, lyocell with incorporated ZnO (CLY), viscose (CV), and flax (LI) fibers were used and the concentration of the ZnO was varied with the concentration of the CLY in the fiber blend. To increase the concentration of ZnO in the samples, the proportion of CLY fibers was increased by decreasing the proportion of CV at a constant content of LI in the blends. The reason for using these fiber blends was that CV/LI blends are very popular in the manufacturing of summer clothing, for which antimicrobial protection is of great importance. The antibacterial activity of samples was estimated against *Escherichia coli* as the Gram-negative and *Staphylococcus aureus* as the Gram-positive bacterium. An important goal of our research was also to determine the influence of LI fibers on the antimicrobial properties of fiber blends. Namely, according to the literature [14,15,16], LI fibers have a reputation for being intrinsically antibacterial. The biodegradation of the samples was estimated by the soil burial test. It was hypothesized that the antimicrobial activity of ZnO in the soil would be impaired, which would beneficially affect the biodegradation process of the fiber blends. 

## 2. Materials and Methods 

### 2.1. Materials

Three types of cellulose fibers were used, i.e., antimicrobial lyocell with incorporated ZnO (CLY, Smart Fiber AG, Rudolstadt, Germany), viscose (CV, Lenzing, Austria), and flax (LI, Liniere de Saint Martin, France). In CLY production, 16% of ZnO was embedded into the fibers by the solvent spinning technique.

For the antibacterial tests, seven samples of three-component CLY/CV/LI fiber blends with a mass of 1.0 ± 0.1 g were prepared with an increased weight fraction of CLY fibers (0%, 15%, 20%, 25%, 30%, 55%, and 70%) along with the appropriate decrease in the weight fraction of CV fibers and a constant amount of LI fibers (30%). Furthermore, one sample was prepared of a two-component fiber blend composed of CLY and CV. The samples were prepared by weighing the individual fibers and assembling them into blends. Single CLY, CV, and LI fiber samples of the same mass were also prepared for the investigation. The sample codes and their compositions are summarized in Table 1.

For the soil burial test, five samples named K (control), CLY(25), CLY(70), CLY(100), CV(100), and LI(100) were selected. The mass of the samples was 15.0 ± 0.1 g. The samples were prepared by weighing the individual fibers and mixing them together with a stirrer (Shirley trash analyzer, Ikon Industries, India) to achieve a homogeneous mixture. The nonwoven fabric samples were put into models with dimensions of 24 cm × 21 cm and clamped between wooden plates with the help of clamps. After 7 days, the clamps were removed, and the compressed samples were wrapped in plastic net. 

### 2.2. Methods

#### 2.2.1. Inductively Coupled Plasma Mass Spectroscopy 

The amount of ZnO in the CLY(100) sample was determined by inductively coupled plasma mass spectroscopy (ICP-MS) using a Perkin Elmer SCIED Elan DRC spectrophotometer. A sample of 0.5 g was prepared in the Milestone microwave system using acid decomposition with 60% nitric acid and 30% hydrogen peroxide. Three measurements were carried out, and the ZnO concentrations are reported as the mean values and standard deviation (SD).

#### 2.2.2. Scanning Electron Microscopy with Energy-Dispersive X-ray Spectroscopy

Scanning electron microscopy (SEM) was performed on a JEOL JSM 6060 LV SEM instrument operating with a primary electron beam of 10 kV accelerating voltage and a working distance of 17 mm. The samples were coated with a thin layer of gold before observation to increase the clarity of the images. SEM analysis was also performed using an SEM Ultra+ microscope (Zeiss, Oberkochen, Germany) equipped with an energy-dispersive X-ray spectrometer (EDS) X-Max 50 (Oxford, UK). The samples were coated with a thin layer of platinum before observation. Sample inspection was carried out with a low acceleration primary electron beam (1 kV at 40 pA) with the detection of secondary electrons.

#### 2.2.3. Fourier Transform-Infrared (FT-IR) Spectroscopy

Fourier transform-infrared (FT-IR) spectra were obtained on a Spectrum GX I spectrophotometer (Perkin Elmer, Great Britain) equipped with an attenuated total reflection (ATR) cell and a diamond crystal (n = 2.0). The spectra were recorded over a range of 4000 to 600 cm^−1^ using 32 scans at a resolution of 4 cm^−1^.

#### 2.2.4. Antibacterial Activity 

The antibacterial activity of samples was estimated for the Gram-negative bacterium *Escherichia coli* ATCC 11229 and the Gram-positive bacterium *Staphylococcus aureus* ATCC 25923 according to the ASTM E 2149–01 standard method [17]. A fabric sample of 1 g was immersed in 50 mL of an NB (nutrient broth, Oxoid, CM0067, Basingstoke, UK) medium of known bacterial concentration in a flask, which was then shaken using a wrist-action shaker. To provide efficient leaching of ZnO from the CLY fibers into the NB culture, 2 h of contact time was selected. Afterwards, viable bacteria were quantified with plate count method using NA (nutrient agar, Biolife, 4018102, Milano, Italy) and incubation of NA plates at 37 °C for 24 h. Two parallel assessments were performed for each fabric sample. The number of colonies was counted, and if the duplicate counts of any sample did not agree within 15%, we discarded that sample and repeated the test. Afterwards, the number of colonies was converted to the number of bacteria as colony-forming units per mL (CFU/mL). The reduction in the number of bacteria, *R*, was calculated as follows: (1)$$R=\frac{\left(B-A\right)}{B}\times 100\;\left(\%\right)$$
where *R* is the bacterial reduction, *A* is the number of bacteria in a flask containing a sample with incorporated antimicrobial CLY fibers after 2 h of contact time, and *B* is the number of bacteria in a flask containing a reference sample without antimicrobial CLY fibers after 2 h of contact time. The *R* value is reported as the mean value and standard error.

To determine the number of viable bacterial cells in the NB medium before and after the specified contact time, the number of bacteria at “0” contact time and after 2 and 24 h of contact with fibers was determined. To determine the number of adhered bacteria, the flasks containing the fibers in contact for 24 h were previously treated in the Sonis 10GT ultrasonic apparatus (Iskra PIO d.o.o., Slovenia) for 10 min (30 kHz, 1100 W) [18]. For comparison, the number of bacteria in the flask containing the inoculum only was determined after the specified contact time. 

#### 2.2.5. Soil Burial Test

The soil burial test was carried out in accordance with the ISO 11721-1:2001 and ISO 11721:2003 standards [19,20]. Following this standard procedure, the container was filled with commercial grade compost. The water content of the soil was 60 ± 5% of its maximum moisture retention capacity. This was held constant during the experiment by spraying with water. The pH of the soil was between 4.0 and 7.5. Fabric samples were buried in the soil for periods of 10, 20, 30, and 50 days. After the defined incubation time, the samples were removed from the test soil, rinsed with running tap water and immersed in 70% ethanol for 30 min before air drying.

#### 2.2.6. Soil Contamination

To investigate the potential impact of buried samples on the soil contamination, soil that was in direct contact with samples K, CLY(25), and CLY(70) during 50 days of burial was transferred into plastic pots. Soil that was not in contact with the fabric samples was used as a reference. Fifty seeds of *Sinapis arvensis* were planted into each pot, and the sprouts were counted after 14 days of growth. Four parallel assessments were performed for each sample, and the mean value and standard deviation were determined. The influence of soil on sprout growth was determined by analysis of variance (ANOVA, F test, 1% level of probability).

## 3. Results and Discussion

### 3.1. Characterization of CLY Fibers 

The SEM images of CLY fibers in Figure 1a,b show the surface morphological properties as well as the distribution of ZnO particles incorporated into the fibers. It was observed that ZnO particles of irregular shapes and particle sizes ranging from 70 to 800 nm were evenly distributed on the CLY fibers. The EDS spectrum (Figure 1c) clearly revealed a characteristic peak at 1.03 keV, which was attributed to Zn.

According to the ICP-MS analysis, the concentration of ZnO in the CLY(100) sample was 120 ± 1 mg/g. 

### 3.2. Antibacterial Activity

The results of the antibacterial activity are presented in Figure 2. It was observed that, irrespective of the content of the CLY component in the fiber blends, antimicrobial activity was more efficient against *E. coli* than against *S. aureus*. Sufficient antibacterial activity, with *R* equal to 86%, was obtained for the CLY(25) sample, and it slightly increased when the CLY content was increased from 25% to 70% in sample CLY(70). This suggests bacteriostatic activity against *E. coli*. On the other hand, low bacterial reduction against *S. aureus* was obtained for all studied samples irrespective of the CLY content, which was unexpected, as it is generally believed that ZnO is more effective against Gram-positive than against Gram-negative bacteria [21,22,23,24]. The results were not in accordance with those from the literature, where, despite higher sensitivity of *E. coli* than *S. aureus* to ZnO, a significant antibacterial protection for both Gram-positive and Gram-negative bacteria was observed for different ZnO-functionalized synthetic fibers [25,26]. Similarly, excellent bacterial reduction of *E. coli* and *S. aureus* was also reported for ZnO/Ag composite functional polyester nonwoven fabric [27]. 

To investigate the reason for the results obtained in the case of the studied cellulose fiber blends, the antimicrobial activity of single CLY fibers (sample CLY(100)) as well as their blend with CV fibers were investigated and compared to the results obtained for the three-component CLY/CV/LI blended samples (Figure 3). As seen in Figure 3, single CLY fibers exhibited high antimicrobial activity against both *E. coli* and *S. aureus* bacteria, and the bacterial reduction against *S. aureus* was expectedly higher than that against *E. coli*. The bacterial reductions caused by CLY(30)—a sample composed of 30% CLY and 70% CV fibers—were very similar to those obtained for the single CLY fibers, indicating the excellent antimicrobial activity of ZnO even at lower CLY content. Surprisingly, the antibacterial activity against *S. aureus* drastically decreased when LI fibers were included into the fiber blends (samples CLY(30) and CLY(70)), regardless of the CLY weight fraction. According to this, it can be suggested that the LI fibers in the fiber blends represented a secondary source of nutrients, promoting the bacteria growth and, consequently, impairing the antimicrobial performance of ZnO. These results are the opposite to those from the literature, in which antibacterial activity of flax fibers was achieved as a result of the action of phenolic compounds in lignin [14,15,16]. 

To gain insight into our results, the differences between the numbers of viable bacterial cells after 2 h of incubation and at contact time “0”, without and in the presence of fiber blended samples, were determined for both bacteria, and the results are presented in Figure 4. It was found that the numbers of both bacteria increased after incubation in the inoculum without any sample and in the presence of sample K, which contained no CLY fibers. Compared to the inoculum itself, the presence of sample K provided much more favorable conditions for the growth of *S. aureus* than for that of *E. coli*. In spite of the fact that the replacement of CV fibers by antimicrobial CLY fibers in the CLY(55) and CLY(70) samples decreased the Δ*N* value for *S. aureus* compared to that obtained in inoculum alone, this could not prevent the bacterial growth during incubation (Figure 4b). Definitely, LI fibers strongly promoted the growth of *S. aureus*, and therefore the concentration of ZnO leached from CLY fibers was too low to provide sufficient antibacterial activity, even at 70% of CLY fibers in the sample. This phenomenon was not observed for *E. coli*, where the ZnO leached from CLY fibers significantly reduced the number of viable bacteria cells during incubation (Figure 4a). 

To verify our findings, the bacterial growth in the presence of single LI fibers (sample LI(100)) was investigated, and the results are summarized in Table 2 and Figure 5. As seen in Table 2, the number of bacteria increased with increased incubation time for both bacteria. Whereas *E. coli* had no tendency for adhering onto LI fibers, a strong adhesion ability was noticed for *S. aureus*. SEM images of LI fibers after incubation revealed that only rare individual cells of *E. coli* were present on the fiber surface (Figure 5a), while cell clusters protected by biofilm (Figure 5c,d) were observed beside individual cells of *S. aureus* (Figure 5b) in many places, confirming the suitability of LI fibers as growth media. To determine the proper number of viable bacterial cells, fiber treatment with ultrasound was necessary. This should be taken into account if standard tests based on the colony count are used.

### 3.3. Biodegradation of Fiber Blends

The influence of CLY fibers in the blended samples on their biodegradation was studied by a soil burial test. In this respect, the following three fiber blended samples were taken into consideration for various reasons: the CLY(25) sample, for which sufficient antibacterial activity was obtained for *E. coli*, the CLY(70) sample, which had the highest content of CLY fibers, and the K sample without CLY fibers as a control. Photos of the samples removed from the test soil after different incubation times are presented in Figure 6. Since the microorganisms in the soil caused fiber rotting, which was reflected in the samples’ browning of the fiber surface, the intensity of the color change could represent a measure of the biodegradation rate. Furthermore, the progression of biodegradation was also reflected in the formation of holes in the samples followed by the tearing of samples into pieces.

A comparison of the photos revealed that the presence of ZnO in the CLY(25) and CLY(70) samples slowed down the biodegradation process compared to the K sample, but it did not prevent the fibers’ biodegradation, even at the highest concentration in the CLY(70) sample. This finding indicated that the biodegradability of the studied samples was maintained. The reason for this could be explained by the fact that the antimicrobial activity of ZnO is photocatalytic, driven by the formation of reactive oxygen species under UV radiation, and that this mechanism of action was not present during the samples’ burial [13]. At the same time, ZnO exhibits poor activity against fungi [28], which represent the majority of soil microorganisms. These results are similar to those obtained for ZnO-coated cellulose fibers [29] but are very different from those obtained for the AgCl-functionalized cellulose fibers, where Ag cations that released from the fiber’s surface exhibited high biocidal activity [30]. 

The growth of microorganisms on the studied samples during biodegradation in soil was inspected by FT-IR spectroscopy (Figure 7). According to our previous study [31], important information about the progress of the biodegradation process can be obtained from the intensity of the bands at 1640 and 1548 cm^−1^, belonging to the amides I and II groups of secondary polyamides [32], resulting from the presence of protein produced by microbial growth on fibers. 

It is seen from the ATR spectra of the CLY(25) sample (Figure 7) that the intensity of the bands at 1640 and 1548 cm^−1^ significantly increased with increased periods of soil burial. Furthermore, both bands were also clearly exposed in the spectrum of CLY(70) after 50 days of soil burial (Figure 8a), confirming the microbial growth on this fiber blend despite the high concentration of ZnO. The progression of biodegradation was also reflected in the decreased intensities of bands in the 1200–900 cm^−1^ spectral region due to structural changes in the cellulose fingerprint. Simultaneously, a band at 1735 cm^−1^ due to C=O stretching in aldehyde groups appeared as a result of cellulose chain degradation [32].

Figure 8 shows the results of the growth of *Sinapis arvensis* sprouts in soil, which was or was not in contact with the K, CLY(25), or CLY(70) samples during 50 days of burial. In none of the pots did all 50 seeds sprout (Figure 8b). This was also true for the seeds that were grown in the reference soil. Although a slightly lower mean value of sprout number was obtained for soil in contact with CLY(70) in comparison to the other samples, according to ANOVA, the soil did not have a significant influence on the sprout growth. This suggested that the rotting of samples and the leaching of ZnO from the samples during the soil burial did not significantly affect the growth conditions for the sprouts, which, consequently, confirmed the sustainability of the fiber blends.

## 4. Conclusions

This study highlights the influence of ZnO on the antibacterial activity and biodegradability of fiber blends composed of CLY, CV, and LI fibers of different blend ratios. The results show that the fiber composition in the samples strongly influenced their antibacterial activity. Whereas CLY fibers blended with CV exhibited high antibacterial activity against both *E. coli* and *S. aureus*, the presence of LI fibers in the three-component blends drastically hindered their antibacterial activity against *S. aureus*. Therefore, LI fibers did not exhibit antimicrobial properties, but, in contrast, they even promoted bacterial growth and, consequently, impaired the antimicrobial performance of ZnO. This phenomenon was accompanied by high adhesion of *S. aureus* onto the LI fibers and the formation of a biofilm. 

The presence of ZnO slowed down the biodegradation process of the fiber blends compared to the sample that did not include CLY fibers, but ZnO did not eliminate the fibers’ biodegradation, even at the highest concentration. This was confirmed by the increased intensity of the absorption bands belonging to the amides I and II groups of secondary polyamides due to the protein portion, indicating the microbial growth on the fibers. Furthermore, the rotting of fiber blends and the leaching of ZnO from the samples during soil burial did not significantly contaminate the soil, proving the sustainability of the fiber blends.

## Figures and Tables

**Figure 1 materials-12-03399-f001:**
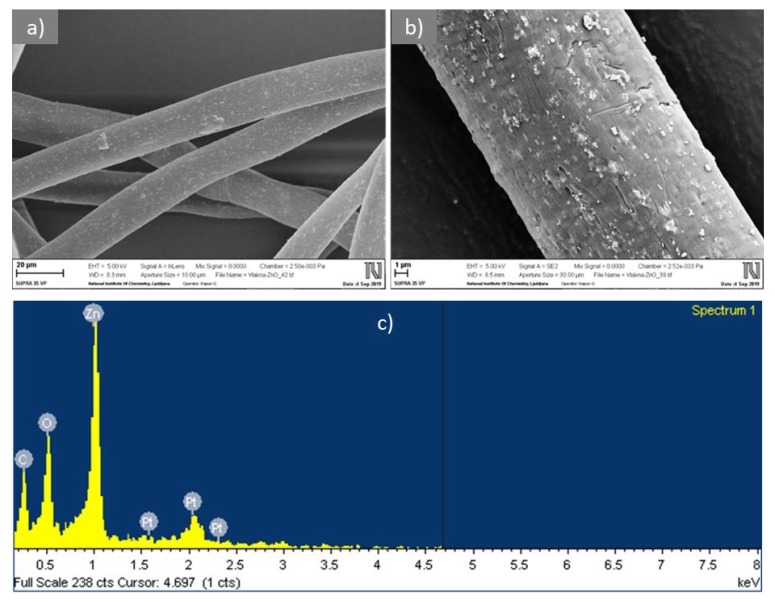
Scanning electron microscopy/Backscattered-Electron images (SEM/BSE) of CLY fibers at lower (**a**) and higher (**b**) magnification and energy-dispersive X-ray spectrometer (EDS) spectrum acquired from ZnO particles (**c**).

**Figure 2 materials-12-03399-f002:**
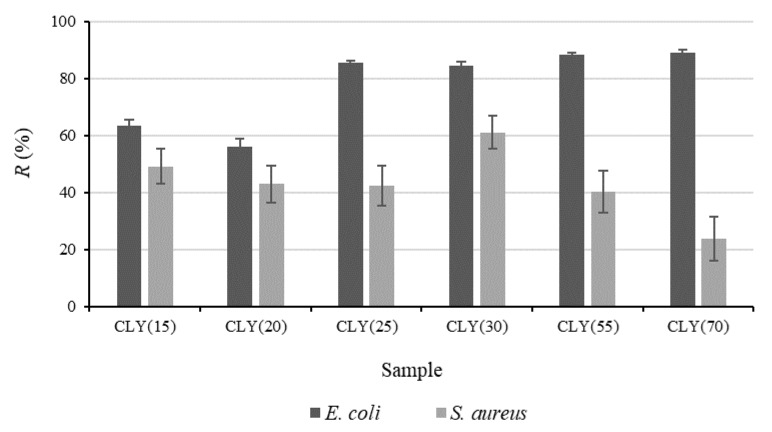
Reduction, *R*, of *Escherichia coli* and *Staphylococcus aureus* versus the mass fraction of CLY fibers, *w*_CLY_, in the fiber blend samples CLY(15) to CLY(70). The K sample was used as a reference.

**Figure 3 materials-12-03399-f003:**
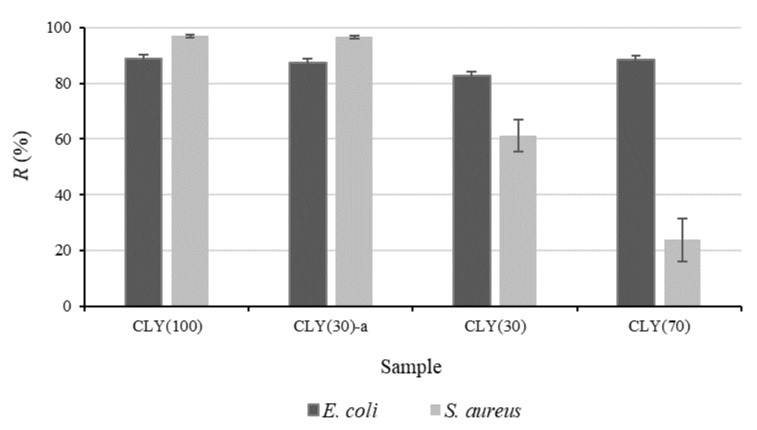
Reduction, *R*, of *E. coli* and *S. aureus* for single CLY fibers (sample CLY(100)) and the fiber blended samples. The K sample was used as a reference for the CLY(30) and CLY(70) samples and the CV(100) sample was used as a reference for the CLY(100) and CLY(30)-a samples.

**Figure 4 materials-12-03399-f004:**
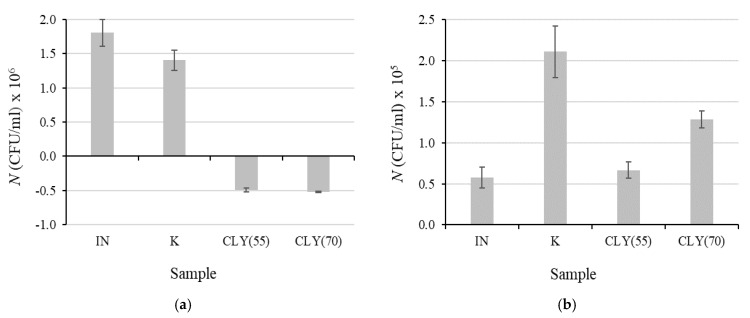
Difference between the numbers of viable bacteria cells, Δ*N*, after and before 2 h of incubation for *E. coli* (**a**) and *S. aureus* (**b**) in inoculum (IN) alone and in the presence of samples K, CLY(55) or CLY(70). The numbers on the x axis denote the number of the CLY sample.

**Figure 5 materials-12-03399-f005:**
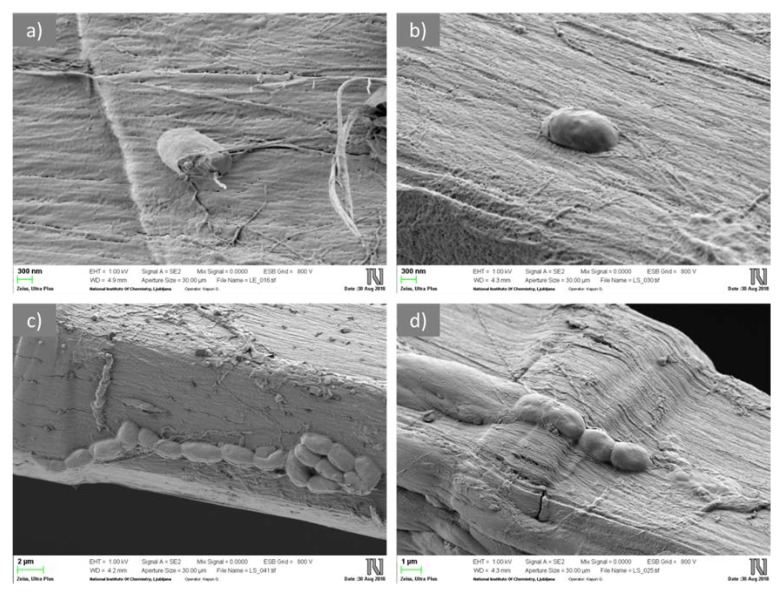
SEM images of *E. coli* (**a**) and *S. aureus* (**b**–**d**) on the surface of LI fibers.

**Figure 6 materials-12-03399-f006:**
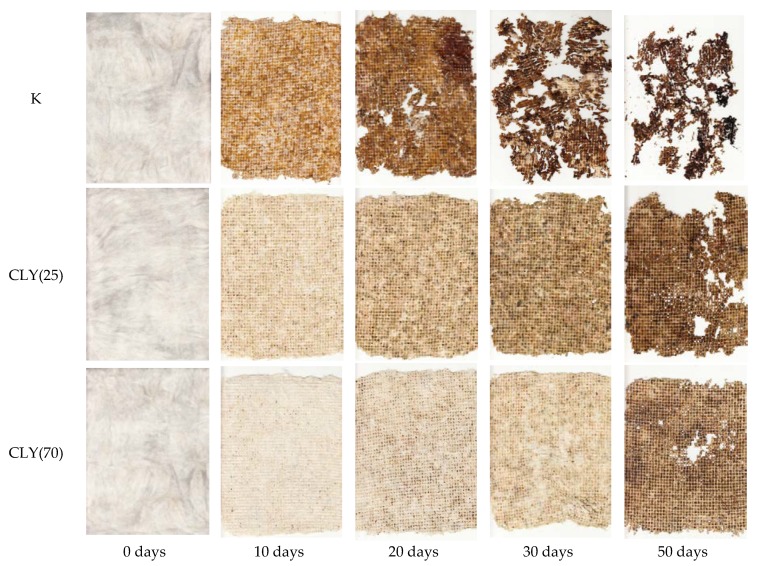
Photos of the K, CLY(25), and CLY(70) samples before (0 days) and after 10, 20, 30, and 50 days of soil burial.

**Figure 7 materials-12-03399-f007:**
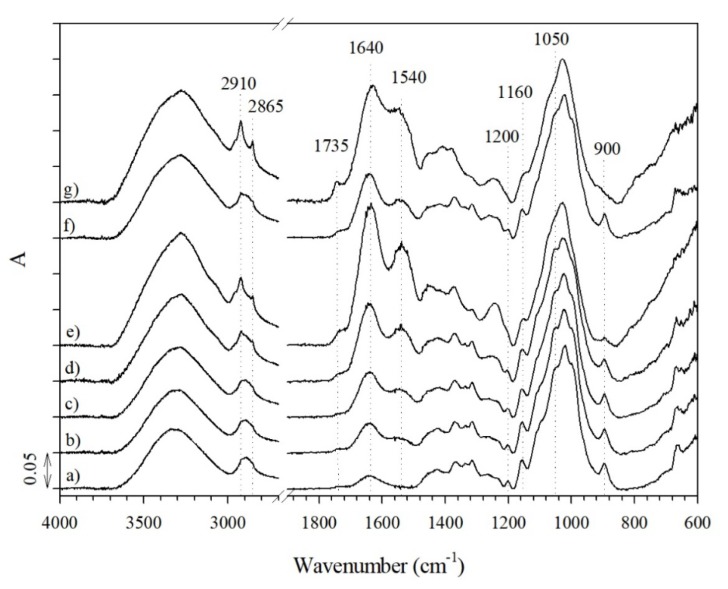
Attenuated total reflection (ATR) spectra of the CLY(25) sample before (**a**) and after 10 (**b**), 20 (**c**), 30 (**d**), and 50 (**e**) days of soil burial, as well as the CLY(70) (**f**) and K (**g**) samples after 50 days of soil burial.

**Figure 8 materials-12-03399-f008:**
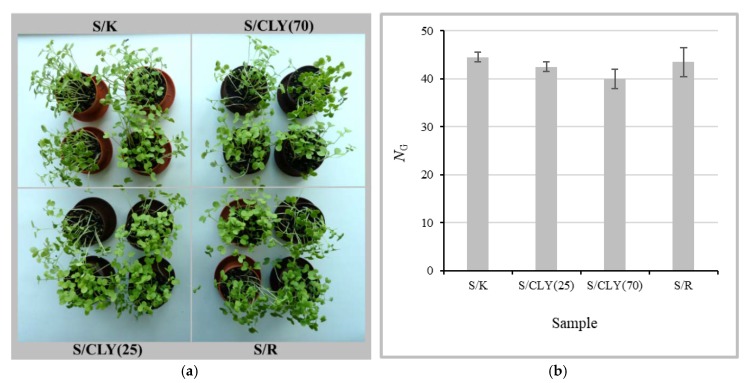
Growth of *Sinapis arvensis* sprouts in soil, which was in contact with samples K (S/K sample), CLY(25) (S/CLY(25) sample), and CLY(70) (S/CLY(70) sample) during 50 days of burial and in the reference soil (S/R sample) with no sample contact (**a**) and number, *N*_G_, of sprouts grown in the soil in 14 days (**b**).

**Table 1 materials-12-03399-t001:** Sample codes and blend ratios.

Sample Code	*w*_CLY_ (%)	*w*_CV_ (%)	*w*_LI_ (%)
K	0	70	30
CLY(15)	15	55	30
CLY(20)	20	50	30
CLY(25)	25	45	30
CLY(30)	30	40	30
CLY(55)	55	15	30
CLY(70)	70	0	30
CLY(30)-a	30	70	0
CLY(100)	100	0	0
CV(100)	0	100	0
LI(100)	0	0	100

Legend: CLY: lyocell with incorporated ZnO, CV: viscose, and LI: flax.

**Table 2 materials-12-03399-t002:** Number of viable bacteria cells, *N*, for *E. coli* and *S. aureus* incubated in the presence of the LI(100) sample for different periods of time, *t*, as well as before and after 10 min of ultrasound treatment.

Bacterium	*N* (CFU/mL)			
	*t* = 0 h	*t* = 2 h	*t* = 24 h, before Ultrasound	*t* = 24 h, after Ultrasound
*E. coli*	6.58 × 10^5^	2.30 × 10^6^	7.75 × 10^7^	7.29 × 10^7^
*S. aureus*	2.63 × 10^5^	3.32 × 10^5^	3.90 × 10^4^	1.96 × 10^7^

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
