# Peer review of "Antibacterial Activity and Biodegradation of Cellulose Fiber Blends with Incorporated ZnO"

_materials, 2019, doi:10.3390/ma12203399_

Round 1
Reviewer 1 Report
The article is well written but before to be published, the authors must solve a problem: write the technique of functionalization the textile material with ZnO.
Author Response
The article is well written but before to be published, the authors must solve a problem: write the technique of functionalization the textile material with ZnO.
Respond: Thank you for your review and constructive opinion. The technique of functionalization the textile material with ZnO was written in the section 2.1 Materials.
Reviewer 2 Report
The aims of the present research was to investigate the influence of ZnO on the antibacterial activity of cellulose fibers as well as their biodegradation.
The manuscript is well written, contain new information and extremely nicely presented, the terminology is highly written, it can be accepted after minor revision. That reviewer did not find the method for the ultrasound treatment, nor the time, type of sonicator, frequency etc.
Congratulation!
Author Response
The aims of the present research were to investigate the influence of ZnO on the antibacterial activity of cellulose fibers as well as their biodegradation.
The manuscript is well written, contain new information and extremely nicely presented, the terminology is highly written, it can be accepted after minor revision. That reviewer did not find the method for the ultrasound treatment, nor the time, type of sonicator, frequency etc.
Respond: Thank you for your review and constructive opinion. The procedure of ultrasound treatment was described in the section 2.2.4.
Reviewer 3 Report
The draft describes the behavior of mixtures in cellulose fibers whose zinc oxide content varies with respect to its action against bacteria and in their biodegradation. In general, manuscript is quite difficult to fill into scope of Material journal because of its specificity of cellulose material. The introduction and bibliography is adequate. As for biodegradation, there is nothing to object to and the results are well understood while the term of reduction R (of antibacterial activity) is ambiguous and tends not to facilitate the understanding of the results. although some ideas underlie that are not well understood by the composition of the mixtures. Among which the following stand out:
1-Excel charts (Figure 2-4) leave much to be desired, they are under worked. If Figure 2 is difficult to understand and Figure 4 directly is not understood.
2- Why is a constant amount of LI (flax) used and fixed in the mixtures if there is a clear effect of this component?
3- In this sense and given the importance of figure 2 because CLY100 and K. are not represented.
4- In Figure 2, it is not explained why there is that effect of lowering up and down again, at least it is not clear, if the percentage of LI is kept constant. If the effect were zinc oxide, why not represent it on this parameter ([Zn2+])?
5- Why in table 1 CV (100) and LI (100) appear? Any results?
6- Only a sample of ZnO is measured in CLY100, and yet in the methodology it seems that all CLY fibers were measured. In addition, it would not be easier to express as 120 ± 1g / kg or 120 ± 1mg / g instead of 120,000 ± 1000mg / kg.
Author Response
The draft describes the behavior of mixtures in cellulose fibers whose zinc oxide content varies with respect to its action against bacteria and in their biodegradation. In general, manuscript is quite difficult to fill into scope of Material journal because of its specificity of cellulose material. The introduction and bibliography is adequate. As for biodegradation, there is nothing to object to and the results are well understood while the term of reduction R (of antibacterial activity) is ambiguous and tends not to facilitate the understanding of the results. although some ideas underlie that are not well understood by the composition of the mixtures.
Respond: Thank you for your review and constructive opinions which significantly improved our article. We clarified in the text that R is reduction in the number of bacteria (i.e. bacterial growth) and not reduction of antibacterial activity. In textiles, antibacterial activity is expressed in the term of R, which is measured according to standardized microbiological test.
1-Excel charts (Figure 2-4) leave much to be desired, they are under worked. If Figure 2 is difficult to understand and Figure 4 directly is not understood.
Respond: Figures 2 and 4 were redrawn to make them understandable.
2- Why is a constant amount of LI (flax) used and fixed in the mixtures if there is a clear effect of this component?
and
3- In this sense and given the importance of figure 2 because CLY100 and K. are not represented.
Respond: The fiber blend composed of 30% of flax and 70% of CV represents a classic fiber blend that is used for a high quality women's and men's clothing. Therefore, the aim of our research was to fabricate antibacterial textile substrate by replacing the viscose fibers with the antibacterial lyocell fibers while maintaining a ratio between the regenerated cellulose fibers (CV and CLY) and flax (LI). Accordingly, an antibacterial activity of fiber blends including flax was presented in Figure 2. The K sample was used as a reference sample B, for which the number of bacteria in a flask was determined to calculate the value of R from Equation 1. It was clarified in the title of Figures 2 and 3.
4- In Figure 2, it is not explained why there is that effect of lowering up and down again, at least it is not clear, if the percentage of LI is kept constant. If the effect were zinc oxide, why not represent it on this parameter ([Zn2+])?
Respond: In Figure 2, we changed the quantity on the x axis to be in accordance with Table 1 and to clarify that R is calculated for different samples. According to the standard method, the R value of the sample is calculated with respect to the reference K and is directly affected by the bacterial growth on the reference. Since the bacterial growth is not always the same on the reference sample K, some deviations could occur, when the number of bacteria increased during the incubation on both, the sample and the reference (Figure 4b). We can not influence it because the bacteria are living matter. These results encouraged us to examine the bacterial growth in more detail.
5- Why in table 1 CV (100) and LI (100) appear? Any results?
Respond: Thank you for your question. Accordingly, we clarified in the title of Figure 3 that CV(100) sample was used as a reference sample for the CLY(100) and CLY(30)-a samples (without flax) to be able to determine the bacterial reduction from equation 1. The LI(100) sample was used for determination of the number of viable bacteria cells presented in Table 2. We clarified this fact in the text (in red).
6- Only a sample of ZnO is measured in CLY100, and yet in the methodology it seems that all CLY fibers were measured. In addition, it would not be easier to express as 120 ± 1g / kg or 120 ± 1mg / g instead of 120,000 ± 1000mg / kg.
Respond: We clarify in the methodology (Section 2.2.1.) that the amount of ZnO was measured in CLY(100) sample and not in CLY fibers.
Round 2
Reviewer 3 Report
I would have expected from the authors that at least the figures in Excel were a little more elaborate. Really with very little they have substantially changed the meaning of work.
Author Response
Respond: Thank you for your review and constructive opinion. The figures were redrawn to be more elaborate. The manuscript was edited by MDPI (Certificate is attached).
